# Population Pharmacokinetic and Pharmacodynamic Analysis for Maximizing the Effectiveness of Ceftobiprole in the Treatment of Severe Methicillin-Resistant Staphylococcal Infections

**DOI:** 10.3390/microorganisms11122964

**Published:** 2023-12-12

**Authors:** Pier Giorgio Cojutti, Simone Giuliano, Renato Pascale, Jacopo Angelini, Carlo Tascini, Pierluigi Viale, Federico Pea

**Affiliations:** 1Department of Medical and Surgical Sciences, Alma Mater Studiorum-University of Bologna, 40138 Bologna, Italy; renato.pascale2@unibo.it (R.P.); pierluigi.viale@unibo.it (P.V.); federico.pea@unibo.it (F.P.); 2Clinical Pharmacology Unit, IRCCS Azienda Ospedaliero-Universitaria di Bologna, 40138 Bologna, Italy; 3Infectious Diseases Clinic, Santa Maria della Misericordia University Hospital of Udine, ASUFC, 33100 Udine, Italy; simone.giuliano@asufc.sanita.fvg.it (S.G.); carlo.tascini@uniud.it (C.T.); 4Infectious Diseases Unit, Department for Integrated Infectious Risk Management, IRCCS Azienda Ospedaliero-Universitaria di Bologna, 40138 Bologna, Italy; 5Institute of Clinical Pharmacology, Santa Maria della Misericordia University Hospital of Udine, ASUFC, 33100 Udine, Italy; jacopo.angelini@asufc.sanita.fvg.it

**Keywords:** ceftobiprole, population pharmacokinetics, MRSA infection, MRSE infection

## Abstract

Ceftobiprole is a fifth-generation cephalosporin used for different Gram-positive bacterial infections. A population pharmacokinetic analysis was conducted in real-life clinical patients to assess the adequacy of current dosages. Population pharmacokinetics was conducted using non-linear mixed effect modeling. Monte Carlo simulations were performed to determine the probability of target attainment (PTA) of free trough or steady-state concentration over MIC (*f*C_trough_/MIC or *f*C_ss_/MIC) ≥ 1 or ≥4 associated with both the standard and intensified dosing regimens adjusted for renal function. Cumulative fraction of response (CFR) against methicillin-resistant *Staphylococcus aureus* (MRSA) and *Staphylococcus epidermidis* (MRSE) were also calculated. A total of 132 patients with 503 concentrations were included. Most of them (107/132, 81.1%) had hospital- or community-acquired pneumonia, endocarditis, and bacteremia. A three-compartment model adequately fitted ceftobiprole concentration-time data. Estimated glomerular filtration rate significantly affected drug clearance. Monte Carlo simulations showed that the optimal target of *f*C_trough_/MIC or *f*C_ss_/MIC ≥ 4 is achieved only with the use of the standard dosages administered by continuous infusion (CI) against MRSA infections in patients with preserved renal function. Intensified dosages administered by CI are needed in patients with impaired renal function and/or augmented renal clearance against MRSA and in patients with preserved renal functions against MRSE.

## 1. Introduction

Ceftobiprole is a fifth-generation cephalosporin with a broad spectrum of antimicrobial activity including primarily Gram-positive cocci, such as penicillin-resistant pneumococci, methicillin-resistant *Staphylococcus aureus* (MRSA), methicillin-resistant *Staphylococcus epidermidis* (MRSE), and *Enterococcus faecalis*. In addition, ceftobiprole is also active against wild type *Enterobacterales* and some strains of *Pseudomonas aeruginosa* [1,2,3]. Ceftobiprole, together with ceftaroline, is a first-in-class beta-lactam with anti-MRSA activity [4].

Ceftobiprole is licensed for the treatment of community-acquired pneumonia (CAP) and of hospital-acquired pneumonia (HAP), excluding ventilator-associated pneumonia (VAP) [5,6,7,8]. Its use is also increasing in several off-label indications, among which bacteremia, infective endocarditis, and bone and joint infections are the most prevalent [4]. Ceftobiprole has good anti-biofilm activity [5] and synergic activity with daptomycin, rifampicin, and vancomycin against MRSA and MRSE [6,7,8].

Like most other beta-lactams, ceftobiprole is hydrophilic and poorly bound to plasma proteins (16%), has a short elimination half-life (3 h) and a negligible potential for drug–drug interactions, and is almost completely eliminated as an unmodified moiety by the renal route [6,9].

Consistently with other cephalosporins, the pharmacokinetic/pharmacodynamic (PK/PD) profile of ceftobiprole features time-dependent antibacterial activity. Pre-clinical models showed that maintenance of concentrations for 30 to 60% of the dosing interval above the MIC (t > MIC) may ensure a 2- to 3-log decrease of MRSA bacterial load over 24 h [10,11]. Post hoc analysis of a clinical trial involving 781 subjects showed that 62.2% t > MIC was associated with microbiological eradication [12].

However, this PK/PD target may be insufficient when treating severe MRSA infections, and more aggressive PK/PD targets up to 100% t > 4–8 × MIC are recommended nowadays for maximizing treatment with 24-continuous infusion (CI) beta-lactams among critically ill patients [13,14,15]. Attaining aggressive PK/PD target could be especially challenging in patients with augmented renal clearance (ARC), a condition occurring in at least one-third of critically ill patients, which may fasten renal elimination of ceftobiprole [16]. Additionally, it may be extremely relevant when dealing with difficult-to-treat infections. In this regard, a recent phase 3 randomized, double-blind, double-dummy, noninferiority trial showed that an intensified ceftobiprole dosing regimen (500 mg every 6 h for 8 days and every 8 h thereafter) was noninferior to daptomycin with respect to overall treatment success in patients with complicated *S. aureus* bacteremia [17].

The aim of this study was to conduct a population PK/PD analysis for testing how standard and intensified dosing regimens may maximize the effectiveness of ceftobiprole in the treatment of severe methicillin-resistant staphylococcal infections.

## 2. Materials and Methods

### 2.1. Study Design

This retrospective study was carried out among adult patients with suspected or documented Gram-positive infections and who were treated with ceftobiprole in the period January 2018 and December 2022. The included patients were those who underwent therapeutic drug monitoring (TDM) of ceftobiprole and who were admitted in two Italian tertiary university hospitals (namely, the IRCCS Azienda Ospedaliero Universitaria di Bologna and the Azienda Sanitaria Universitaria Friuli Centrale of Udine). The study was approved by the local Ethics Committees of both hospitals. According to hospital policies pertaining to retrospective observational investigations, patient’s signed informed consent to participate to this study was waived.

Ceftobiprole was used in mono- or combo-therapy. The starting dosages adjusted for classes of estimated glomerular filtration rate (eGFR) were 500 mg q8h over 3 h extended infusions (EI) if eGFR ≥ 50 mL/min/1.73 m^2^, 500 mg q12h over 3 h-EI if eGFR 30–50 mL/min/1.73 m^2^, and 250 mg q12 h over 3 h-EI if eGFR < 30 mL/min/1.73 m^2^.

After at least 2 days from starting therapy, patients underwent real-time TDM coupled with expert clinical pharmacological advice (ECPA) for dose adjustments. TDM-based ECPA was aimed at attaining an optimal PK/PD target, defined as a steady-state plasma trough concentrations (C_trough_) to MIC ratio of 4–8 (corresponding to a 100% t > 4–8 × MIC). In case of empirical treatment, the MIC value was set at the EUCAST clinical breakpoint against staphylococci, namely 2 mg/L. In both centers, TDM of ceftobiprole was routinely performed 5 days weekly, Monday to Friday, measuring C_trough_. Whenever feasible, additional blood samples were collected at standardized timepoints, namely at the end and 1 h after the 3 h infusion. Blood samples were analyzed within 3 h from their delivery to the lab, within the same day of collection. TDM results and ECPA were made available through the hospital intranet system within the mid-afternoon of the same day.

Demographic data (age, gender, weight, and height) and clinical data (type and site of infection, bacterial clinical isolate and MIC for ceftobiprole, serum creatinine, eGFR, ceftobiprole daily dose, and eventual co-treatments with other antibiotics) were retrieved from each patient’s medical record. The CKD-EPI formula [18] was used to estimate patient eGFR. Microbiological and/or clinical outcome and adverse events related to ceftobiprole were also collected. Patient clinical outcome was defined at end of treatment as cured in case of complete resolution of signs and symptoms of infection along with a consistent reduction of inflammatory biomarkers (C-reactive protein and procalcitonin) and/or microbiological eradication. Clinical failure was defined as failed in case of lack of clinical response or worsening of patient clinical conditions despite ceftobiprole treatment.

### 2.2. Drug Analysis

Ceftobiprole was analyzed by means of a validated high-performance liquid chromatography tandem mass spectrometry (LC-MS/MS) method [19]. Briefly, after blood sample centrifugation for 10 min at 9000× *g* and addition of methanol solution, an aliquot of the clear supernatant was transferred to an autosampler vial and a volume of 10 μL was injected into the LC-MS/MS system. Chromatography was performed by means of an Agilent 1290 Infinity II UHPLC. The analysis of the samples was carried out on a ZORBAX Eclipse plus C18 column, (2.1 × 50 mm, 1.8 μm particle size; Agilent, Santa Clara, CA, USA), and chromatographic separation was conducted at 25 °C. The calibration range was set from 0.5 to 100 mg/L, and three quality controls (QCs) were used and set at 3.5 (low QC), 35 (medium QC), and 70 (high QC) mg/L. A solution of cefiderocol-d12 10 mg/L in methanol was used as an internal standard. Intraday precision (mean CV%) and accuracy (mean bias%) met the EMA requirements Specifically, the intra- and inter-day coefficients of variation ranged from 9.0% to 9.6% and from 5.2% to 9.7%, respectively.

The lower limit of quantification (LLOQ) of ceftobiprole was 0.5 mg/L, corresponding to the lowest point of the calibration curve.

### 2.3. Population Pharmacokinetic Modelling

Plasma concentration-time profiles of ceftobiprole were analyzed by means of nonlinear mixed-effects modeling using the stochastic approximation maximization (SAEM) algorithm implemented within the Monolix software (version 2023R1, Lixofit, Antony, France).

Initially, a base model without covariates was developed by comparing one-, two-, and three-compartment pharmacokinetic models with zero-order administration and first-order elimination from the central compartment. All individual parameters were considered to be log-normally distributed. Between-subject variability was described through exponential random effects. Correlations among random effects were evaluated in the variance-covariance matrix and applied into the structural model accordingly.

Different error models (constant, proportional or combined) were tested for describing the residual variability. At this stage, the selection of the most appropriate model was based on a reduction of both the objective function value (OFV) of greater than 3.84-point and the Akaike Information Criteria (AIC) of greater than 2-points, and on the lowest relative standard error (RSE) of the population pharmacokinetic estimates.

Once the base model was defined, the following covariates were tested on the pharmacokinetic parameters: age, gender, weight, height, and eGFR. The relationship between parameters and covariates was modeled as proportional or additive function for binary covariates and as a power function for continuous covariates. Covariate selection was carried out according to a forward/backward process. In the forward step, the inclusion of a covariate in the model was based on the result of the Pearson’s correlation test between each covariate and the random effect on estimated pharmacokinetic parameter. In the backward step, the Wald test was used to test whether any covariate could be removed from the full covariate model.

Model evaluations were based on the evaluation of standard goodness of fit plots. These were the observed vs. population- and individual-predicted concentrations, the distribution of the weighted residuals vs. time and vs. predicted concentrations, and the visual predictive check (VPC) plot. The VPC plot compares graphically the empirical percentiles (summarized by the 10th, 50th, and 90th percentiles of the observed data) with the theoretical percentiles computed using multiple Monte Carlo simulations (summarized by colored areas). Model performance was also tested by assessing the distribution of the normalized prediction distribution errors (NPDE). The 95% confidence interval of each parameter in the final model was simulated from 1000 nonparametric bootstraps based on resampling by means of the Rsmlx (R speaks Monolix) package of R (Version 2023.1.5).

### 2.4. Monte Carlo Simulation and Probability of Target Attainment

By using the final pharmacokinetic model, 1000 Monte Carlo simulations were performed for both licensed and intensified ceftobiprole dosages, administered either by EI or by CI, in five different classes of renal function. The licensed dosages were as follows: 250 mg q12h over 2 h-EI or 500 mg q24h CI for eGFR < 30 mL/min/1.73 m^2^; 500 mg q12h over 2 h-EI or 1000 mg q24h CI for eGFR 30–50 mL/min/1.73 m^2^; 500 mg q8h over 2 h-EI or 1500 mg q24h CI for eGFR 51–80 mL/min/1.73 m^2^ and for eGFR 81–130 mL/min/1.73 m^2^; 500 mg q8h over 2 h-EI or 1500 mg q24h CI for eGFR > 130 mL/min/1.73 m^2^. The intensified dosages were: 250 mg q8h over 2 h-EI or 750 mg q24h CI for eGFR < 30 mL/min/1.73 m^2^; 500 mg q8h over 2 h-EI or 1500 mg q24h CI for eGFR 30–50 mL/min/1.73 m^2^; 500 mg q6h over 2 h-EI or 2000 mg q24h CI for eGFR 51–80 mL/min/1.73 m^2^, for eGFR 81–130 mL/min/1.73 m^2^ and for eGFR > 130 mL/min/1.73 m^2^; 500 mg q4h over 2 h-EI or 3000 mg q24h CI for eGFR > 130 mL/min/1.73 m^2^.

Likelihood of PK/PD target attainment (PTA), in terms of free (*f*) C_trough_/MIC ratio for EI and of *f*C_ss_/MIC ratio for CI, was calculated at 72 h and defined as optimal when ≥4 and quasi-optimal when ≥1. Free ceftobiprole concentrations were calculated by assuming a 16% protein binding, as previously showed [6]. PTAs ≥ 90%were considered optimal.

The cumulative fractions of response (CFRs) against the MIC distribution of methicillin-resistant *Staphylococcus aureus* (MRSA) and methicillin-resistant *Staphylococcus epidermidis* (MRSE) as reported by the EUCAST [20], were assessed for each of the tested ceftobiprole dosing regimens. CFRs ≥ 90% were considered as optimal.

## 3. Results

### 3.1. Study Population

Overall, 132 patients were included in the study. Demographic and clinical characteristics are reported in Table 1. Median (min-max) age, weight, and eGFR were 71 (61.8–79) years, 73.5 (65–89) kg, and 83.7 (50.5–101.7) mL/min/1.73 m^2^, respectively. Ceftobiprole was most frequently used for treating hospital-acquired pneumonia (28.8%, 38/132), endocarditis (20.5%, 27/132), primary bloodstream infections (16.6%, 22/132), and community-acquired pneumonia (15.2%, 20/132).

The etiological agent of the infection was identified in 60.6% (80/132) of patients. A single bacterial Gram-positive infection was observed in 81.3% of these (65/80) (11.3% (9/80) due to MRSA, 10% (8/80) due to MRSE, 22.5% (18/80) due to MSSA, 2.5% (2/80) due to MSSE, and 28.8% (23/80) due to Enterococci). Infections were polymicrobial in 18.8% (15/80) of patients.

Ceftobiprole was used in mono-therapy in 33.3% cases (44/132) and in combo-therapy in 66.7% (88/132) of cases, mainly with daptomycin (30.7%; 27/88), ampicillin 27.3%; 24/88), or fosfomycin (9.1%; 8/88).

Adverse effects probably related to ceftobiprole use were observed in four patients (two cases of seizures and two cases of measles-like rash).

### 3.2. Population Pharmacokinetic Modelling

A total of 503 ceftobiprole plasma concentrations were included in the population pharmacokinetic model. Median (min-max) ceftobiprole C_trough_ was 7.4 (0.7–37.2) mg/L, whereas median concentrations at the end and 1 h after the 3 h infusion were 18.4 (4.6–55.3) and 14.8 (5.0–55.5) mg/L, respectively.

Ceftobiprole pharmacokinetics was best described by a three-compartment model (OFV/AIC being 3168.92/3180.92, 3095.87/3115.87, and 3085.11/3113.11 for the one-, two-, and three-compartment model, respectively). eGFR and gender were the two clinical covariates significantly associated with the pharmacokinetic parameters. After adding these covariates as a power function of ceftobiprole CL and V_1_, respectively, OFV/AIC decreased further to 3022.7/3056.7. Parameter estimates of both the base and the final model are shown in Table 2.

The median (min–max) population values of CL, V_1_, and V_2_ were 4.04 (1.0–13.8) L/h, 19.7 (7.5–83.8) L, and 46.6 (7.6–526.0) L, respectively. Reliability of the estimates of the final model was confirmed by the low values of the relative standard error for all the parameters (excepted for Q_3_ and the random effects on Q_3_) and by the high value of the coefficient of the linear regression of observed vs. individual predicted concentrations (R^2^ = 0.855), as shown in Figure 1.

The visual predictive check (VPC) plot of the final model demonstrated acceptable predictive performance of the data set given that the 10th, 50th, and 90th percentiles of the observed data were within the simulated prediction intervals (Figure 2). The symmetry test for normalized prediction distribution errors (NPDE) showed that residuals were around zero (*p* = 0.368) and normally distributed (*p* = 0.04 at the Shapiro–Wilk test for NPDE).

### 3.3. Monte Carlo Simulation for Estimating PK/PD Target Attainment

Figure 3 and Figure 4 show the PTAs of quasi-optimal and optimal *f*C_trough_/MIC ratio and/or *f*C_ss_/MIC ratio, respectively, with 2 h-EI and/or 24 h-CI administration of the different dosages of ceftobiprole in the five classes of renal function. Looking at the MIC value of 2 mg/L, namely the EUCAST clinical breakpoint against *Staphylococcus aureus*, PTAs ≥ 90% were attainable only for a quasi-optimal PK/PD target with the licensed ceftobiprole dosages administered by EI in the classes of eGFR ranging from 30 to 130 mL/min/1.73 m^2^, and by CI in that of eGFR > 130 mL/min/1.73 m^2^. PTAs ≥ 90% of the optimal PK/PD target were attainable by CI administration of the licensed dosages but only against pathogens with an MIC value up to 1 mg/L in those of eGFR between 30 and 130 mL/min/1.73 m^2^ and to 0.5 mg/L in classes of eGFR < 30 and >130 mL/min/1.73 m^2^.

Table 3 summarizes the CFRs of quasi-optimal and optimal PK/PD targets achievable with the tested dosages of ceftobiprole against the EUCAST distribution of MRSA and MRSE in the different classes of renal function. The use of the licensed dose by CI granted ≥90% attainment only of quasi-optimal PK/PD targets in all the classes of renal function against both MRSA and MRSE. However, when considering optimal PK/PD target attainment, CFRs ≥ 90% were granted only against MRSA by using CI administration of the licensed dosages in the classes of eGFR between 30 and 130 mL/min/1.73 m^2^, and of intensified dosages of 500 mg q4h in the class of eGFR ≥ 130 mL/min/1.73 m^2^. Conversely, the same dosing approach granted against MRSE CFRs ≥ 90% of only quasi-optimal PK/PD targets. The intensified dosages allowed optimal PK/PD target attainment against MRSA in the class of eGFR < 30 mL/min/1.73 m^2^ and against MRSE in the classes of eGFR between 30–130 mL/min/1.73 m^2^.

## 4. Discussion

The present population pharmacokinetic study of ceftobiprole was the first based on a multi-centric cohort of real-life patients with different types of Gram-positive infections. The findings highlighted that the licensed dosages administered by EI may be insufficient for attaining optimal PK/PD targets of efficacy against staphylococcal infections, especially among patients without renal dysfunction who have MRSA infections and those affected by MRSE infections.

Our real-life popPK model showed that creatinine clearance was a major covariate associated with ceftobiprole CL, in agreement with the results previously observed by four population pharmacokinetic studies that were carried out among adult patients coming from phase I–III registrative clinical trials [19,20,21,22]. Indeed, the population estimates of our model were higher for V_1_ (19.7 L) and lower for CL (4.04 L/h) compared to those observed previously. Lodise et al. [21] developed a three-compartment pharmacokinetic model based on 150 subjects (39 patients with SSTI and 111 healthy volunteers), and showed that population V_1_ and CL estimates of ceftobiprole were 7.65 L and 4.8 L/h, respectively. Kimko et al. [22] performed another three-compartment population pharmacokinetic model including a total of 595 subjects (433 patients with SSTI and 162 healthy volunteers), based on both rich and sparse blood sampling, and found V_1_ and CL estimates of 7.4 L and 5.36 L/h, respectively. Muller et al., in another three-compartment population pharmacokinetic study concerning a mixed population of 171 subjects made up of healthy volunteers, critically ill patients, patients with SSTI, and patients with nosocomial pneumonia [23], showed that V_1_ and CL estimates were 15.5 L and 4.74 L/h. Finally, the same author in another model [24] including 200 patients of Asiatic and non-Asiatic ethnicity with SSTI and nosocomial pneumonia found population V_1_ and CL estimates of 15.4 L and 7.1 L/h.

The higher V_1_ estimates in our study group might be explained by the severity of acute illness leading to capillary leakage in most cases (>80% of patients with severe acute infections), differing from what was observed in the subjects included in the other studies with less severe type of infections (namely SSTI) or who were healthy volunteers. The lower ceftobiprole CL may be explained by the lower eGFR estimates observed in our cohort (83.7 mL/min/1.73 m^2^) than reported in three of the aforementioned studies (median CL_CR_ between 106 and 111 mL/min) [21,23,24].

From a PK/PD perspective, Monte Carlo simulations have pinpointed that the current licensed dosages of ceftobiprole administered by EI may be adequate only for attaining quasi-optimal PK/PD target when dealing with MRSA infections in all of the classes of eGFR except ARC. Conversely, this is not the case when dealing with MRSE infections or for attaining optimal PK/PD target in both MRSA and MRSE infections.

A paradigm shift in the concept of desired PK/PD target attainment with beta-lactam is currently occurring nowadays [25] from the traditional 50–100% t > MIC up to the aggressive PK/PD target of 100% t > 4–8 × MIC, which was considered optimal since it was recently shown to minimize the risks of microbiological failure and of resistance development [13,14,16,26]. Unfortunately, attainment of this aggressive target is never checked during drug development as it is not a requirement of the regulatory authorities. In regard to ceftobiprole, Muller et al. assessed the PTA of 100% T > MIC among patients treated for SSTI or nosocomial pneumonia, and Monte Carlo simulation showed that values >90% were achievable only against pathogens with an MIC up to 1 mg/L, although no data about simulated dosing regimens and tested classes of eGFR were reported [24].

The present Monte Carlo analysis showed that administering the licensed dosages by CI rather that by EI may be helpful for attaining optimal PK/PD target in some of the eGFR classes when dealing with MRSA infections (namely 30–50, 51–80, and 81–130 mL/min/1.73 m^2^), but not when dealing with MRSE infections. Conversely, the intensified dosages, especially when administered by CI, may represent an appropriate approach for dealing with MRSA infections in all of the eGFR classes and with MRSE infections in most of the e GFR classes (except eGFR < 30 or >130 mL/min/1.73 m^2^).

These latter PK/PD findings may support the results of the recent phase 3 randomized, double-blind, double-dummy, noninferiority trial showing that, in patients with complicated *S. aureus* bacteremia, an intensified ceftobiprole dosing regimen of 500 mg every 6 h by EI for 8 days and every 8 h by EI thereafter was noninferior to daptomycin with respect to overall treatment success [17]. Since, in that study, the proportion of *S. aureus* microbiological eradication was 82%, it could be speculated that administering the intensified doses by CI rather than by EI would have allowed even higher microbiological eradication rates, especially among patients who could greatly benefit from this approach, namely those without renal dysfunction.

We acknowledge some limitations of our study. The retrospective study design should be recognized. Moreover, whereas CFR analysis against MRSA was supported by an EUCAST strain collection accounting for more than 15,000 clinical isolates, that against MRSE was significantly smaller (only 185 clinical isolates).

## 5. Conclusions

In conclusion, the present study was the first to assess the population pharmacokinetics of ceftobiprole in a large cohort of patients with different staphylococcal infections. More aggressive PK/PD target attainment such as 100% t > 4–8 × MIC (optimal PK/PD target) should be pursued for the treatment of severe *S. aureus* infections. The use of standard dosages administered by CI is required against MRSA infections in patients with preserved renal function, whereas intensified dosages administered by CI are needed in patients with impaired renal function and ARC against MRSA and in patients with preserved renal functions against MRSE.

## Figures and Tables

**Figure 1 microorganisms-11-02964-f001:**
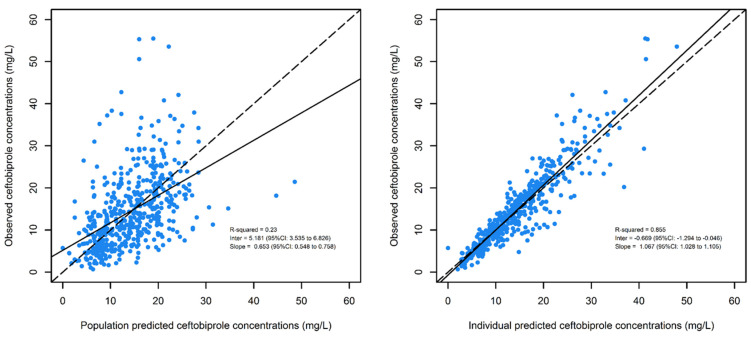
Diagnostic scatter plots of population pharmacokinetic model. (**Left panel**): observed versus. population-predicted concentrations. (**Right panel)**: observed versus individual-predicted concentrations.

**Figure 2 microorganisms-11-02964-f002:**
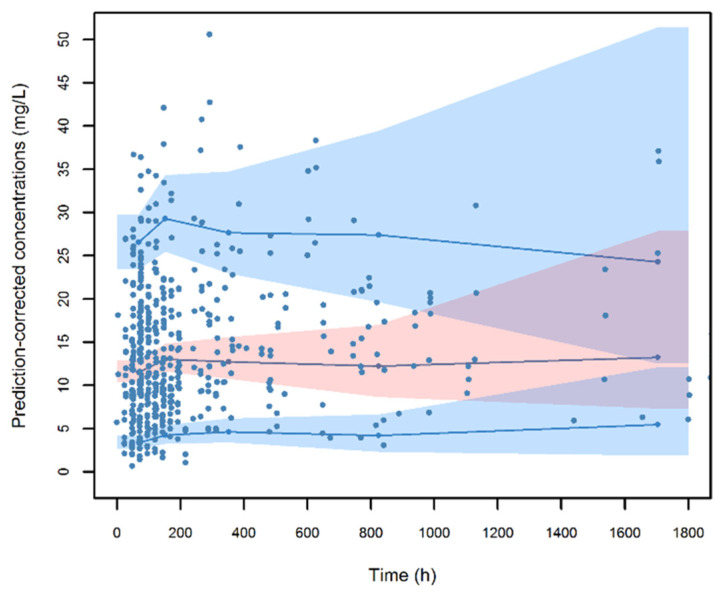
Visual predictive check (VPC) of ceftobiprole plasma concentration vs. time for the final model. The continuous lines indicate the 10th, 50th, and 90th percentiles of observed data, while the shaded areas represent 90% prediction intervals from the corresponding percentiles calculated from simulated data.

**Figure 3 microorganisms-11-02964-f003:**
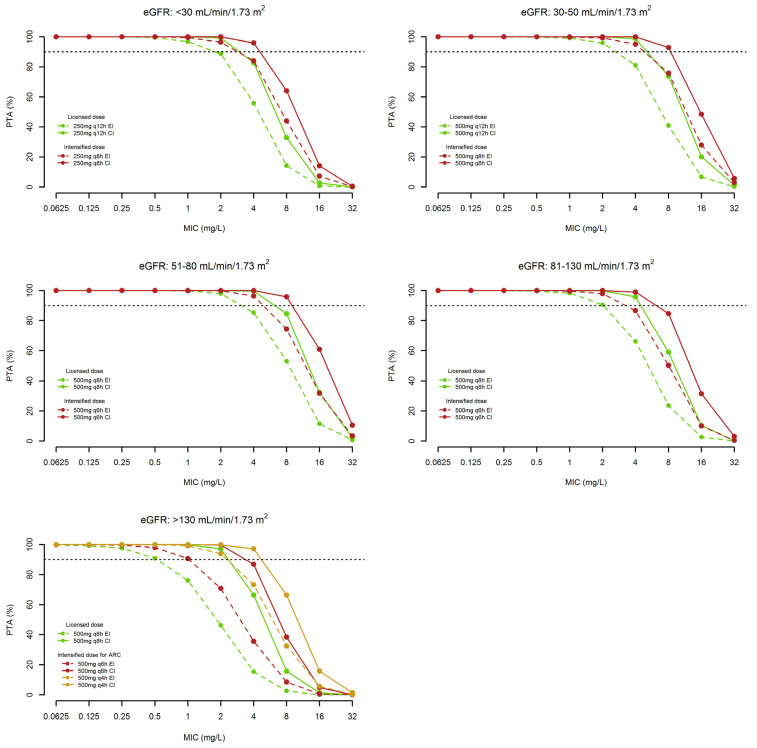
Probability of target attainment (PTA) of quasi-optimal PK/PD target (namely *f*C_trough_/MIC or *f*C_ss_ /MIC ≥ 1) with different ceftobiprole dosages in relation to five different classes of renal function against MRSA with different MIC distribution. Horizontal dotted line identifies PTA ≥ 90%.

**Figure 4 microorganisms-11-02964-f004:**
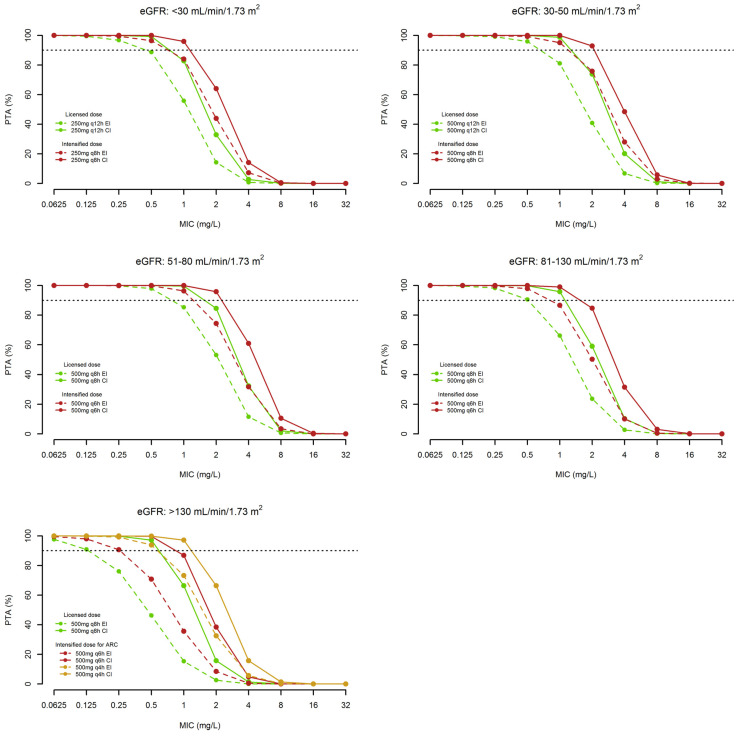
Probability of target attainment (PTA) of optimal PK/PD target (namely *f*C_trough_/MIC or *f*C_ss_ /MIC ≥ 4) with different ceftobiprole dosages in relation to five different classes of renal function against MRSA with different MIC distributions. Horizontal dotted line identifies PTA ≥ 90%.

**Table 1 microorganisms-11-02964-t001:** Demographic and clinical characteristics of patients (n = 132).

Characteristic		Value
Age (years)		71.0 (61.8–79.0)
Gender (male/female)		86/46
Body weight (kg)		73.5 (65.0–89.0)
BMI (kg/m^2^)		25.7 (22.5–30.1)
Serum creatinine (mg/dL)		0.90 (0.68–1.36)
eGFR (mL/min/1.73 m^2^)		83.7 (50.5–101.7)
Serum albumin (g/L)		3.1 (2.7–3.4)
Type of infection, n (%)		
	Hospital-acquired pneumonia	38 (28.8)
	Endocarditis	27 (20.5)
	Bloodstream infection	22 (16.6)
	Community-acquired pneumonia	20 (15.2)
	Bone and joint infections	9 (6.8)
	Skin and soft tissue infections	9 (6.8)
	Device-related infections	4 (3.0)
	CNS infections	3 (2.3)
Patients with identified microbiological isolates, n (%)		80 (60.6)
Ceftobiprole treatment		
	Median dose (mg daily)	1500 (1000–1500)
	Trough concentration (mg/L)	7.6 (4.9–11.7)
	Length of treatment (days)	10.0 (2.0–81.0)
	Patients with co-administered antibiotics, n (%)	88 (66.7)
Treatment outcome in assessable patients (n = 126)		
	N. of patients with microbiological eradication	118 (96.7)
	N. of patients with clinical cure	88 (69.8)

Data are presented as median (IQR) for continuous variables and as number (%) for dichotomous variables. BMI, body mass index; CNS, central nervous system; eGFR, estimated glomerular filtration rate.

**Table 2 microorganisms-11-02964-t002:** Population parameter estimates of the base and final models, and bootstrap results based on 1000 simulations.

Parameter		Base Model Estimate (% RSE)	Covariate Model Estimate (% RSE)	Bootstrap Median (95% CI)
Typical values				
	CL (L/h)	3.43 (7.33)	1.7 (11.9)	1.63 (1.48–1.76)
	V_1_ (L)	21.29 (13.5)	14.77 (19.7)	14.12 (14.37–16.72)
	Q_2_ (L/h)	7.12 (19.0)	6.11 (13.6)	6.54 (5.89–7.61)
	V_2_ (L)	35.79 (44.2)	46.16 (34.8)	46.56 (38.75–50.76)
	Q_3_ (L/h)	4.04 (31.5)	42.41 (61.6)	44.71 (31.57–67.84)
	V_3_ (L)	4.08 (26.9)	4.76 (13.9)	4.74 (4.30–6.00)
Covariate effect				
	eGFR on CL	-	0.011 (12.6)	0.01 (0.010–0.012)
	Gender on V_1_	-	0.39 (54.8)	0.38 (0.29–0.49)
Inter-individual variability				
	on CL (%CV)	61.9 (8.6)	47.4 (8.43)	47.38 (43.92–50.90)
	on V_1_ (%CV)	68.5 (29.6)	75.3 (19.4)	79.52 (50.91–93.09)
	on Q_2_ (%CV)	29.6 (46.5)	35.0 (34.5)	21.23 (16.10–37.19)
	on V_2_ (%CV)	717.4 (21.2)	268.1 (19.4)	477.17 (224.12–762.68)
	on Q_3_ (%CV)	80.9 (37.6)	235.2 (45.1)	123.01 (58.18–360.44)
	on V_3_ (%CV)	50.9 (40.2)	30.7 (30.0)	26.44 (18.14–31.76)
Correlation				
	CL and V_1_	-	0.6 (24.8)	0.52 (0.40–0.65)
Residual variability				
	a	1.43 (12.0)	1.44 (13.4)	1.49 (1.07–1.65)
	b	0.22 (6.8)	0.22 (5.8)	0.23 (0.21–0.25)

a and b represent additive and proportional residual error model, respectively.

**Table 3 microorganisms-11-02964-t003:** Cumulative fraction of response (CFRs) of quasi-optimal (namely *f*C_trough_/MIC or *f*C_ss_/MIC ratio ≥ 1) and optimal (namely *f*C_trough_/MIC or *f*C_ss_/MIC ratio ≥ 4) PK/PD targets achievable with the tested dosages of ceftobiprole against the EUCAST MIC distribution of MRSA and MRSE in the different classes of renal function.

Ceftobiprole Dosages and Classes of eGFR	MRSA		MRSE	
	Quasi-Optimal PK/PD Target	Optimal PK/PD Target	Quasi-Optimal PK/PD Target	Optimal PK/PD Target
eGFR < 30 mL/min/1.73 m^2^				
250 q12h EI	95.9	58.2	88.1	52.9
250 q12h CI	99.7	79.1	91.3	73.9
250 q8h EI *	98.8	80.7	90.7	75.2
250 q8h CI *	100	91.4	91.4	85.4
eGFR 30–50 mL/min/1.73 m^2^				
500 q12h EI	98.6	78.4	90.4	72.9
500 q12h CI	99.9	94.5	91.4	88.1
500 q8h EI *	99.7	92.6	91.2	85.8
500 q8h CI *	100	98.4	91.4	90.7
eGFR 51–80 mL/min/1.73 m^2^				
500 q8h EI	99.5	83.2	91.1	77.1
500 q8h CI	100	96.8	91.4	89.7
500 q6h EI *	99.4	93.4	91.3	86.6
500 q6h CI *	100	98.9	91.4	91.0
eGFR 81–130 mL/min/1.73 m^2^				
500 q8h EI	97.2	66.0	89.4	60.8
500 q8h CI	100	90.5	91.4	84.8
500 q6h EI *	99.4	83.5	91.0	77.7
500 q6h CI *	100	96.5	91.4	90.0
eGFR > 130 mL/min/1.73 m^2^				
500 q8h EI	75.2	22.2	69.3	18.3
500 q8h CI	99.2	66.7	91.1	61.3
500 q6h EI *	89.2	41.1	82.3	36.2
500 q6h CI *	99.8	82.3	91.2	77.1
500 q4h EI *	98.4	72.2	90.3	66.7
500 q4h CI *	100	92.5	91.4	86.4

* Intensified dosages tested in different classes of eGFR. CI, 24 h-continuous infusion; EI, 2 h-extended infusion.

## Data Availability

The data presented in this study are available on request from the corresponding author.

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
