# Peer review of "Population Pharmacokinetic and Pharmacodynamic Analysis for Maximizing the Effectiveness of Ceftobiprole in the Treatment of Severe Methicillin-Resistant Staphylococcal Infections"

_microorganisms, 2023, doi:10.3390/microorganisms11122964_

Round 1
Reviewer 1 Report
Comments and Suggestions for Authors
See attached file

Author Response
The present study describes population pharmacokinetics of ceftobiprole in a large cohort of patients comparing standard versus intensified doses in either extended or continuous infusion. According to the study results, optimal PK/PD target is achieved only with the use of standard doses in CI for MRSA infections in all patients except those patients with augmented renal clearance in which the use of intensified doses in CI are needed. For patients with MRSE infections, even intensified doses in CI are not enough to attain optimal PK/PD in patients with eGFR < 30 ml/min or those with eGFR>130 ml/h.
I find this work very interesting and well written, and I just have some comments:
R.: We thank the reviewer for appreciating our work
(i) Lines 98-99: The authors explained that two additional blood samples are collected in addition to Ctrough sample. Why do you use these two? In order to calculated exposure?AUC? Please explain.
R.: We thank the reviewer for this question. TDM-based dose adjustments of ceftobiprole was based on Ctrough in all the patients. However, whenever clinically feasible, additional sampling within the dosing interval were also requested and for practical reasons these were standardized at the reported timepoints, namely at the end of infusion and at 1h after the infusion. This information was added in the Methods.
(ii) Lines 108-111: A think that a more detailed explanation about clinical outcomes should be needed. At what time do clinical cure was measured? What biomarkers were employed?
R.: We thank the reviewer for this question. Clinical cure was measured at end of treatment. The inflammatory biomarkers used for assessing patient response to treatment were C-RP and procalcitonin. This information was added in the Methods.
(iii) I would suggest calculating the PTAs of optimal fCtrough/MIC ratio and or fCss/MIC ratio for eGFR < 30 ml/min and eGFR > 130ml/min patients with MRSE infections with EI and CI (even if higher than licensed doses are needed).
R.: For patients with eGFR <30 mL/min/1.73m2 and >130 mL/min/1.73m2 we already showed that higher than licensed doses (250 mg q8h and 500 mg q4h, respectively) were unable to reach optimal target attainment. We think that testing even higher doses could support improper prescription. For this reason we prefer not to simulate higher exposure, especially in patients at risk of drug accumulation.
Minor comments:
Line 27. “acquired infections” → do you mean “acquired pneumonia”?
R.: Corrected. Thanks.
Line 71. “S.aureus” → “S.aureus”
R.: Corrected. Thanks.
Lines 189: data is expressed by median (min-max) but the same data is expressed as median (IQR) in Table 1. I think that IQR is more adequate, please change. Lines 189-193: information is already in the table. Please, don’t repeat information.
R.: We thank the reviewer for this observation. However, we prefer to report both information, as the min-max range informs on the extremes boundaries of the distribution, whereas the IQR informs on how the data are distributed around the value of tendency.
Lines 201-203: information is already in the table. You should choose either the text or the table for this information.
R.: We agree with referee’s suggestion. Accordingly, we removed from the text the information already present in table 1.
Table 1: please define BMI and CNS
R.: Thanks. Definition of BMI and CNS were added in the footnotes of the table.
Line 303: “infctions” → “infections”
R.: Corrected. Thanks.
Line 325: “from what observed” → “from what was observed”
R.: Corrected. Thanks.
Reviewer 2 Report
Comments and Suggestions for Authors
The paper is well written and results are interesting. The cohort is quite large (132 patients), however the patients with a microbiological isolate are only 65 and the CFRs against the MIC distribution was evaluated only in 9 cases with a MRSA infections and 8 patients with MRSE. Thus, wider cohort are needed to confirm the results.
Anyway, I think that the paper is of scientific interest and add new data on the management of MRSA infections especially in patients with altered renal function.
Author Response
The paper is well written and results are interesting. The cohort is quite large (132 patients), however the patients with a microbiological isolate are only 65 and the CFRs against the MIC distribution was evaluated only in 9 cases with a MRSA infections and 8 patients with MRSE. Thus, wider cohort are needed to confirm the results.
Anyway, I think that the paper is of scientific interest and add new data on the management of MRSA infections especially in patients with altered renal function.
R.: We thank the referee for appreciating our work. However, CFR calculation was based on the entire EUCAST MIC distribution and not on the observed patients. The EUCAST MIC distribution accounts of more than 15.000 clinical isolates for MRSA, but only 185 for MRSE. This discrepancy was highlighted in the discussion as a limit in our evaluation.
Reviewer 3 Report
Comments and Suggestions for Authors
I have reviewed the article title " Population pharmacokinetic and pharmacodynamic analysis for maximizing the effectiveness of ceftobiprole in the treatment of severe methicillin-resistant staphylococcal infections" and I have found major flaws that need to be revised before further processing.
Title pharmacodynamic? or pharmacodynamics?, please check and correct it
Abstract
Gram-positive infections? Or Gram-positive bacterial infection?, please revise.
The whole abstract section is written unprofessionally. The authors should rewrite a structural, abstract comprising, background, MM, findings, conclusion and recommendation.
Keywords should be revised as: MRSA; MRSE; infections; ceftobiprole; pharmacokinetics
2.1 Study Design
Line 79. Please revise. “This retrospective study was carried out …”
Most of the sentences are too long which contradict their meaning, please split sentences and make them short. Such as, 68-71; 79-82; 87-91’ 92-97, and so on.
3.1. Study population.
Overall, a total of 132…
Either remove the term :overall”, or :a total of”
Line 94. Gram-positive 194 monomicrobial infections…., should be revise to “a single bacterial infection..”, according to the objective of the current study.
Results
Microbiological isolates were identified in 60.6 % (80/132) of patients?. What are microbiological isolates? The whole article is ambiguous and the authors have no specific direction, just jumping here and there. Please focus on the aim of the study and write the whole article in a sequence.
Discussion
Our population pharmacokinetic study… What is our population?,
Our Monte Carlo analysis? What is our Monte Carlo analysis?,
Please revise the whole article carefully and resubmit it.
The authors should focus on the tested organism, there are thousands of Gram-positive bacteria, so the term Gram-positive organisms (as in line 368) seem ambiguous
Comments on the Quality of English LanguageMost of the sentences are too long which contradict their meaning, please split sentences and make them short. Such as, 68-71; 79-82; 87-91’ 92-97, and so on. The article must be revise carefully.
Author Response
I have reviewed the article title " Population pharmacokinetic and pharmacodynamic analysis for maximizing the effectiveness of ceftobiprole in the treatment of severe methicillin-resistant staphylococcal infections" and I have found major flaws that need to be revised before further processing.
Title pharmacodynamic? or pharmacodynamics?, please check and correct it
R.: We thank the referee for this observation. The correct word is “pharmacodynamic”, in the form of an adjective to the word analysis.
Abstract
Gram-positive infections? Or Gram-positive bacterial infection?, please revise.
R.: We thank the referee for this observation. The adjective “bacterial” was added.
The whole abstract section is written unprofessionally. The authors should rewrite a structural, abstract comprising, background, MM, findings, conclusion and recommendation.
R.: We thank the referee for this observation. However, the style of Microorganisms for abstract is a single paragraph structured into background, methods, results and conclusions, but without headings. In particular, we re-wrote the introduction and aims (first three lines of the abstract). Methods are comprised between lines 3-8. Results are comprised between lines 8-14. Conclusion are comprised between lines 14-16.
Keywords should be revised as: MRSA; MRSE; infections; ceftobiprole; pharmacokinetics
R.: We thank the referee for this observation. However, the keyword “population pharmacokinetics” was preferred over “pharmacokinetics” as it better reflects what was done in this study. Moreover, we think that the suggested keyword “infections” may be integrated with MRSA and MRSE.
2.1 Study Design
Line 79. Please revise. “This retrospective study was carried out …”
R.: We thank the referee for this observation, and we re-phrased the sentence accordingly.
Most of the sentences are too long which contradict their meaning, please split sentences and make them short. Such as, 68-71; 79-82; 87-91’ 92-97, and so on.
R.: We thank the referee for suggesting us to shorten the statements in the Methods. Statements were made short.
3.1. Study population.
Overall, a total of 132… Either remove the term :overall”, or :a total of”
R.: We thank the referee for this observation. The term “a total of” was removed from the text.
Line 94. Gram-positive 194 monomicrobial infections…., should be revise to “a single bacterial infection..”, according to the objective of the current study.
R.: We thank the referee for this observation. The statement was rephrased as suggested by the reviewer.
Results
Microbiological isolates were identified in 60.6 % (80/132) of patients?. What are microbiological isolates? The whole article is ambiguous and the authors have no specific direction, just jumping here and there. Please focus on the aim of the study and write the whole article in a sequence.
R.: We thank the referee for this observation. This statement means that in 80 out of 132 patients the etiological agent of the infection was identified. The sentence was re-phrased accordingly.
Discussion
Our population pharmacokinetic study… What is our population?,
R.: We thank the referee for this observation. The term “our” was substituted with the term “the present”
Our Monte Carlo analysis? What is our Monte Carlo analysis?,
R.: We thank the referee for this observation. The term “our” was substituted with the term “the present”
Please revise the whole article carefully and resubmit it. The authors should focus on the tested organism, there are thousands of Gram-positive bacteria, so the term Gram-positive organisms (as in line 368) seem ambiguous
R.: We agree with the referee. Accordingly, we modified the statement by specifying that the infections were staphylococcal infections.
Comments on the Quality of English Language
Most of the sentences are too long which contradict their meaning, please split sentences and make them short. Such as, 68-71; 79-82; 87-91’ 92-97, and so on. The article must be revise carefully.
R.: We thank the referee for suggesting us to shorten the statements.
Reviewer 4 Report
Comments and Suggestions for Authors
In this study, the authors conducted a comprehensive population pharmacokinetic analysis of ceftobiprole in 132 patients with Gram-positive infections, utilizing non-linear mixed-effect modeling. The establishment of a three-compartment model, fitting the concentration-time data, highlighted the significant impact of glomerular filtration rate on drug clearance. Monte Carlo simulations were employed to assess the probability of target attainment for free trough or steady-state concentrations over MIC (fCtrough/MIC or fCss/MIC) ≥1 or ≥4, considering standard and intensified dosing regimens adjusted for renal function. The findings indicate that optimal target attainment for ceftobiprole against MRSA infections necessitates standard dosages administered by continuous infusion in patients with preserved renal function. Conversely, intensified dosages administered by continuous infusion are deemed necessary for patients with impaired renal function and/or augmented renal clearance against MRSA, as well as for those with preserved renal function against MRSE.
Major Concern:
The study is well-designed and holds significant potential for optimizing treatment strategies for MRSA and MRSE-infected patients in the region. However, a major concern arises regarding the mixing of male and female data in Figures 1-4. Given the known gender-specific differences in the pharmacokinetics and pharmacodynamics of ceftobiprole, it is imperative that the authors either provide a clear rationale for this amalgamation or redesign the figures separately for both genders, with corresponding mentions throughout the manuscript.
Minor Concerns:
1. The resolution of the figures requires improvement.
2. It is recommended to include an additional column in the figures indicating the gender for each parameter to enhance clarity and facilitate a more nuanced interpretation of the results.
Author Response
In this study, the authors conducted a comprehensive population pharmacokinetic analysis of ceftobiprole in 132 patients with Gram-positive infections, utilizing non-linear mixed-effect modeling. The establishment of a three-compartment model, fitting the concentration-time data, highlighted the significant impact of glomerular filtration rate on drug clearance. Monte Carlo simulations were employed to assess the probability of target attainment for free trough or steady-state concentrations over MIC (fCtrough/MIC or fCss/MIC) ≥1 or ≥4, considering standard and intensified dosing regimens adjusted for renal function. The findings indicate that optimal target attainment for ceftobiprole against MRSA infections necessitates standard dosages administered by continuous infusion in patients with preserved renal function. Conversely, intensified dosages administered by continuous infusion are deemed necessary for patients with impaired renal function and/or augmented renal clearance against MRSA, as well as for those with preserved renal function against MRSE.
Major Concern:
The study is well-designed and holds significant potential for optimizing treatment strategies for MRSA and MRSE-infected patients in the region. However, a major concern arises regarding the mixing of male and female data in Figures 1-4. Given the known gender-specific differences in the pharmacokinetics and pharmacodynamics of ceftobiprole, it is imperative that the authors either provide a clear rationale for this amalgamation or redesign the figures separately for both genders, with corresponding mentions throughout the manuscript.
R.: We thank the referee for appreciating our work. As far as the role of gender on the pharmacokinetics of ceftobiprole is concerned, it is worth noting that our population pharmacokinetic analysis identified gender only as a covariate on the volume of distribution of the central compartment and not on clearance. This means that gender may affect only the loading dose but not the maintenance dose, for which only eGFR should be considered for dose adjustments. This should be expected, considering that ceftobiprole, being an hydrophilic cephalosporin, is primarily excreted unchanged in the urine, predominantly by glomerular filtration. The non-significant role of gender on ceftobiprole clearance was also reported by Murthy B. et al (Clin Pharmacokinet 2008; 47 (1): 21-33) who reported that, after repeated dosing and adjusting for body weight, there were no apparent differences in the systemic exposure of ceftobiprole in healthy men and women. Therefore, there is no robust reason to separate the population by gender.
Minor Concerns:
- The resolution of the figures requires improvement.
R.: We thanks the referee for this comment. Resolution of the figure was increased at 600 dpi, which is above the required standard (i.e., 300 dpi).
- It is recommended to include an additional column in the figures indicating the gender for each parameter to enhance clarity and facilitate a more nuanced interpretation of the results.
R.: As previously explained, gender was not a significant covariate on drug clearance, which is the PK parameter that drives dosing.
Round 2
Reviewer 3 Report
Comments and Suggestions for Authors
After revising the manuscript contents have been improved and can be accepted.
Reviewer 4 Report
Comments and Suggestions for Authors
I have no more concerns